# Application of Ultrasound Radiomics in Differentiating Benign from Malignant Breast Nodules in Women with Post-Silicone Breast Augmentation

**DOI:** 10.3390/curroncol32010029

**Published:** 2025-01-03

**Authors:** Ling Hao, Yang Chen, Xuejiao Su, Buyun Ma

**Affiliations:** Department of Medical Ultrasound, West China Hospital, Sichuan University, Chengdu 610041, China; lingerlinger@icloud.com (L.H.); sophie-0627@163.com (Y.C.); suxuejiao163@163.com (X.S.)

**Keywords:** ultrasound radiomics, breast nodule, benign and malignant, silicone breast augmentation, machine learning

## Abstract

Purpose: To evaluate the diagnostic value of ultrasound radiomics in distinguishing between benign and malignant breast nodules in women who have undergone silicone breast augmentation. Methods: A retrospective study was conducted of 99 breast nodules detected by ultrasound in 93 women who had undergone silicone breast augmentation. The ultrasound data were collected between 1 January 2006 and 1 September 2023. The nodules were allocated into a training set (*n* = 69) and a validation set (*n* = 30). Regions of interest (ROIs) were manually delineated using 3D Slicer software, and radiomic features were extracted and selected using Python programming. Eight machine learning algorithms were applied to build predictive models, and their performance was assessed using sensitivity, specificity, area under the ROC curve (AUC), accuracy, Brier score, and log loss. Model performance was further evaluated using ROC curves and calibration curves, while clinical utility was assessed via decision curve analysis (DCA). Results: The random forest model exhibited superior performance in differentiating benign from malignant nodules in the validation set, achieving sensitivity of 0.765, specificity of 0.838, and an AUC of 0.787 (95% CI: 0.561–0.960). The model’s accuracy, Brier score, and log loss were 0.796, 0.197, and 0.599, respectively. DCA suggested potential clinical utility of the model. Conclusion: Ultrasound radiomics demonstrates promising diagnostic accuracy in differentiating benign from malignant breast nodules in women with silicone breast prostheses. This approach has the potential to serve as an additional diagnostic tool for patients following silicone breast augmentation.

## 1. Introduction

Breast augmentation surgery has been widely embraced by individuals seeking aesthetic enhancements since its inception. Advances in technology have led to the evolution of breast prosthesis materials, from early options like autologous fat and paraffin to the now widely used silicone-based compounds [1]. Despite ongoing concerns regarding the safety of silicone prostheses—particularly in genetically susceptible populations, where they may trigger chronic inflammatory responses or even lead to breast implant-associated anaplastic large cell lymphoma (BIA-ALCL) [2]—there is currently no conclusive evidence demonstrating harm to human health. International guidelines have been established to standardize the follow-up management of breast augmentation surgeries [3,4].

Ultrasound is a common imaging modality used for monitoring patients post-surgery; however, the presence of silicone prostheses can compromise image quality and reduce diagnostic accuracy. Recent advancements in radiomics—a technique that extracts and analyzes numerous quantitative features from medical images—have shown promise in improving the accuracy of distinguishing between benign and malignant breast nodules [5]. Previous studies have demonstrated the significant potential of radiomics in predicting the classification of breast tumors [6]; however, limited research has focused on radiomic features of breast nodules in the context of silicone breast prostheses.

Given these considerations, this study aimed to evaluate the potential of ultrasound radiomic features in distinguishing benign from malignant breast nodules in women who have undergone silicone breast augmentation. By establishing predictive models, this research seeks to provide safer and more efficient medical services for this specific patient group, thereby contributing to the advancement of precision medicine.

## 2. Materials and Methods

### 2.1. Data Source

All data were collected from the ultrasound examination database of the Department of Ultrasound Medicine at West China Hospital, Sichuan University. Considering the specificity of the study subjects and the potential of a small sample, we set the data collection period from 1 January 2006 to 1 September 2023 to include all cases of women who underwent breast augmentation for cosmetic purposes. We recorded the ultrasound examination information for the target population along with corresponding pathological follow-up findings. Additionally, clinically relevant information about the target group was collected through the hospital management information system (HIS). The information collected included basic information (name, ID, sex, age), ultrasound details (examination date, examination number, ultrasound report description, ultrasound diagnosis results), and clinically relevant information (history of breast augmentation, medical history, pathological results). All data were systematically curated into an Excel table. For patients with multiple nodules, the following prioritization strategy was employed. Firstly, nodules were categorized as benign or malignant based on the pathological results. Secondly, bilateral nodules in the same patient were registered separately for the left and right sides. Finally, for multiple nodules on the same side, only the largest nodule was registered.

### 2.2. Study Subjects

All study subjects were either outpatients or inpatients treated at West China Hospital, Sichuan University. The inclusion criteria were (1) presence of breast prostheses clearly mentioned in the ultrasound examination records, and (2) ultrasound examination records with corresponding pathological follow-up results.

In evaluating the initially collected ultrasound records, we established exclusion criteria based on the assessment of pathological, clinical, and ultrasound information for each record. Specifically, we excluded ultrasound records with pathological follow-up results that did not match our study objective, clinical information that introduced confounding variables, or substandard ultrasound images. The exclusions included: (1) pathological follow-up results limited to tissue within or outside the capsule of breast implants, without diagnostic information on the breast nodules detected by ultrasound; (2) pathological follow-up records with inconsistent or unclear locations compared to the nodule sites identified by ultrasound; (3) patients with a history of breast cancer, radiotherapy, or chemotherapy; (4) records with incomplete clinical information; (5) ultrasound images where lesions were obscured by measurement icons; (6) ultrasound images showing lesions too large to be fully visualized; (7) ultrasound records lacking two-dimensional or color Doppler ultrasound images; (8) duplicate examination records from the same patients (multiple examinations for the same patient).

### 2.3. Outcome Measures

In this study, the outcome measures of the predictive model were based on the pathological diagnostic results of breast nodules, categorized as benign or malignant. Pathological diagnoses were reached according to internationally accepted WHO histological standards. Tissue samples, obtained via needle biopsy or surgical resection, were reviewed and confirmed by specialist pathologists experienced in breast pathology. The pathological results for all final included cases were obtained within six months after the ultrasound examination.

### 2.4. Acquisition of Ultrasound Pictures, Preprocessing, and ROI Delineation

The diagnostic breast ultrasound equipment used in this study included devices manufactured by Philips, GE, Hitachi, Mindray, and Siemens, with probe frequencies ranging from 4 to 15 MHz and color Doppler image scales ranging from ±2.5 to 10 cm/s. All ultrasound images were acquired by physicians with over three years of experience in breast ultrasound examinations, and the images and reports were reviewed by senior physicians.

Due to the retrospective nature of the analysis, the ultrasound images were downloaded from the hospital’s PACS system in JPG format. During the ultrasound image preprocessing phase, we effectively reduced the risk of bias through the anonymization of case numbers and collaborative work between two junior physicians. Specifically, one physician was responsible for assigning each case a random and unique number. Each unique number was used to label different ultrasound image files, ensuring accurate correspondence between the unique numbers, ultrasound images, and clinicopathological information. The same physician annotated the pathology results on the ultrasonographic images that had been delineated with regions of interest (ROIs) for subsequent computer recognition.

Another junior physician delineated the ROIs for all lesions without prior knowledge of the nodules’ pathological results. ROI delineation was performed using 3D Slicer software (version 5.6.1). All ultrasound images were standardized. During the delineation process, the physician used tools such as zooming, dragging, drawing, and tracing to precisely outline the margins of the lesions. For lesions with unclear margins, the delineations were reviewed and corrected by a senior physician. Figure 1 illustrates the ROI segmentation for benign and malignant breast nodules in ultrasound images. The completed ROI masks were saved in nii.gz format, and the standardized ultrasound images were saved in nii format, preparing them for subsequent ultrasound feature extraction in the Python language environment.

### 2.5. Dataset Construction

To ensure the independence and effectiveness of model training and validation, we utilized the random number generation function in SPSS software to assign a unique random number between 0 and 1 to each case. Subsequently, the dataset was divided into a training set and a validation set in a 7:3 ratio. Specifically, the top 70% of cases with the highest random numbers were allocated to the training set, totaling 69 cases, while the remaining 30% of cases were allocated to the validation set, totaling 30 cases.

### 2.6. Feature Extraction

We performed feature extraction on all ultrasound images in both the training set and the validation set. Ultrasound imaging omics features were extracted using the SimpleITK (version 2.3.1) and Radiomics (version 3.0.1) toolkits in the Python programming language. Three different filters were used to preprocess these images: Wavelet, Logarithm, and SquareRoot. Features were extracted from both the original and preprocessed images, resulting in seven categories of features: first-order statistics, shape, gray-level co-occurrence matrix (GLCM), gray-level run-length matrix (GLRLM), gray-level size-zone matrix (GLSZM), neighborhood gray-tone difference matrix (NGTDM), and gray-level dependence matrix (GLDM) [7].

### 2.7. Feature Screening

We used only the data from the training set to screen out the most predictively valuable features. Initially, non-numeric types of features and constant features were removed, as they did not contribute substantially to model training. Next, the correlations between features were calculated, and a correlation threshold of 0.95 was set. By identifying and removing highly correlated features, we reduced the redundancy of the data, thereby enhancing the efficiency and accuracy of model training. Subsequently, the Spearman correlation analysis method was used to assess the degree of association between each feature and the target variable (benign or malignant). Based on the preset correlation threshold (absolute value > 0.45), we selected features that showed relatively significant correlations with the target variable. Finally, using the LASSO algorithm with 5-fold cross-validation, we selected the optimal features with importance greater than 0.05.

### 2.8. Model Establishment and Evaluation

For the binary classification problem, we utilized eight widely recognized machine learning algorithms to build models: random forest, logistic regression, decision tree, gradient boosting, naïve Bayes, support vector machine (SVM), k-nearest neighbor (KNN), and multilayer perceptron (MLP). Each model was trained using the processed training set, while an independent validation set was used to evaluate model performance, ensuring an unbiased assessment.

For each model, we used sensitivity, specificity, and AUC to evaluate its diagnostic efficacy. Additionally, accuracy, Brier score, and log loss score were used to assess the predictive efficacy of the model. ROC curves for the eight models were plotted for both the training set and the validation set to compare their diagnostic efficacy. When comparing the predictive efficacy of the eight models, accuracy focused on the overall correctness of the model’s classification, the Brier score focused on the accuracy of the model’s predicted probabilities, and the log loss score focused on the quality of the model’s predicted probabilities, particularly penalizing cases where the predicted probability of the actual outcome was extremely inaccurate. The best model was determined by comparing the AUC values of the eight models in the validation set. Finally, a calibration curve for the best-performing model was plotted to assess its stability, and decision curve analysis (DCA) was performed to evaluate its clinical utility.

### 2.9. Statistical Analysis

Clinical baseline statistics were conducted for the final included cases, including patient age, nodule pathological type, method of breast augmentation, and duration since augmentation. Comparisons were made regarding the distribution of lesions in terms of breast laterality, quadrant, and maximum diameter between the training set and the validation set. Statistical analysis was performed using SPSS version 23.0 software (IBM Corp., Armonk, NY, USA). The Shapiro–Wilk test was used to determine whether continuous variables conformed to a normal distribution. For continuous variables, comparisons were made using the independent-sample t-test or the Mann–Whitney U test, depending on the distribution. Categorical variables were compared using the chi-squared test or Fisher’s exact test, as appropriate. A *p*-value < 0.05 was considered statistically significant, with a confidence interval set at 95%.

Feature extraction and screening, model training, and validation were conducted using Python 3.9 (Python Software Foundation). Sensitivity, specificity, AUC, accuracy, Brier score, and log loss score were calculated using the scikit-learn and numpy libraries. The ROC curves, calibration curves, and DCA curves were generated using the matplotlib library.

## 3. Results

### 3.1. Clinical and Pathological Information

We initially included 521 ultrasound records. After applying strict exclusion criteria, we ensured that the final cohort consisted exclusively of adult women who had undergone breast augmentation for cosmetic purposes, with all nodules having matched pathological follow-up results and clear ultrasound images. Ultimately, we included a total of 99 nodules from the ultrasound images of 93 patients. The entire dataset construction process is illustrated in Figure 2.

The detailed results of the pathological types of the 99 breast nodules are presented in Table 1. A total of 93 patients were included in our study. Among them, 87 patients had a single lesion in one breast, 4 patients had a single lesion in both breasts, and 2 patients had two lesions in one breast. The age range of the patients was broad, with the youngest being 23 years old and the oldest being 75 years old, and the median age was 46 years. Regarding surgical methods, statistical results showed that 26 patients underwent augmentation with liquid silicone injections, while 67 patients had augmentation using silicone gel implants. The duration since augmentation ranged from as short as 2 months to over 20 years, with the data presented in Table 2 in intervals of 5 years. Notably, 13 patients did not provide the exact duration of their breast augmentation at the time of consultation.

Comparisons were made between the training set and the validation set for the distribution of pathological types (benign, malignant), side of the lesion, quadrant of the lesion, and the maximum diameter of the lesion. The statistical methods and results are shown in Table 3. The results indicate no statistically significant differences in the distributions mentioned above between the two sets. Notably, the distribution of the maximum diameter of the lesions did not conform to a normal distribution (*p* < 0.001).

### 3.2. Feature Extraction and Selection

From each ROI in the training set, we extracted 1374 ultrasound imaging omics features. Initially, we excluded 42 features that were constant or lacked statistical significance. Next, we excluded 575 features with correlations higher than 0.95 to reduce redundancy. Further, using Spearman correlation analysis, we identified 36 features with relatively high correlations (greater than 0.45) with the target variable (benign/malignant). Finally, through LASSO selection, we identified nine optimal features. The feature selection and parameter-tuning process of the LASSO algorithm are shown in Figure 3a,b, and a horizontal bar chart displaying the importance of these optimal features is shown in Figure 3c.

### 3.3. Model Evaluation

The specific values of diagnostic and predictive performance metrics for the eight models in both the training and validation sets are provided in Table 4, and the ROC curves are displayed in Figure 4.

In the training set data, the random forest, decision tree, and gradient boosting algorithms achieved the highest AUC values, sensitivity, specificity, and accuracy. Multilayer perceptron, logistic regression, support vector machine, and k-nearest neighbor also performed relatively well, with AUC values of 0.998, 0.977, 0.959, and 0.947, respectively. In contrast, the naïve Bayes algorithm yielded the lowest AUC value of 0.843.

In the validation set data, the three models with the highest AUC values were: random forest (0.787), k-nearest neighbor (0.717), and logistic regression (0.701). Decision tree performed best in terms of sensitivity (0.778), followed by k-nearest neighbors and random forest (both at 0.765). In terms of specificity, support vector machine and gradient boosting performed best (both at 0.846), followed by random forest (0.838). Regarding predictive performance metrics, random forest achieved the highest accuracy (0.796). All eight models had low Brier scores, indicating little difference between the predicted probabilities and the observed outcomes. However, it is noteworthy that decision tree and multilayer perceptron exhibited higher log loss scores, reaching 10.074 and 9.833, respectively, suggesting larger discrepancies in the predicted probabilities for individual cases. The best-performing model in terms of Brier score and log loss score was random forest, with values of 0.197 and 0.599, respectively.

In the context of predicting the benign or malignant status of breast nodules in the presence of silicone breast prostheses, the random forest model we established achieved the highest AUC value in the validation set. The calibration curve (Figure 5) shows that the predicted probabilities of this model align closely with the actual outcomes. Finally, the DCA curves for the random forest model in both the training and validation sets (Figure 6) indicate its potential clinical utility.

## 4. Discussion

The rise of breast augmentation can be traced back to the 1960s, with experts in aesthetic plastic surgery consistently focusing on the safety, comfort, and durability of breast prostheses, conducting extensive research on their manufacturing processes [8,9,10]. As early as the late 20th century, medical experts raised concerns about the safety of breast prostheses, primarily focusing on issues such as the potential for silicone gel to trigger autoimmune diseases, lymphoma, and breast cancer, which have garnered significant attention over decades of clinical research and investigation [11,12,13]. In recent years, multiple studies have indicated that silicone gel may alter the body’s microenvironment in genetically susceptible individuals, leading to rejection reactions and chronic inflammatory responses [14,15,16]. Additionally, the rupture and infection of breast prostheses may further exacerbate these adverse conditions. Consequently, individual variability and potential risks associated with silicone breast augmentation have led clinicians to place greater emphasis on post-augmentation breast screening. Ultrasound examination, favored for its lack of radiation, convenience, and low cost, is one of the most commonly used screening methods in clinical practice, making it the most acceptable choice for the majority of women who have undergone breast augmentation in China. As a comprehensive tertiary hospital holding a significant position within the healthcare system of southwest China, our institution ensures that cases selected from our patient pool are highly representative.

During the case collection phase, we implemented strict inclusion and exclusion criteria to focus on women who had undergone breast augmentation for cosmetic reasons. Patients with a history of mastectomy and reconstructive surgery or those who had undergone radiotherapy or chemotherapy for breast cancer were excluded, as these factors can alter tissue response patterns in the breast, potentially influencing the high-throughput features observed in ultrasound images. However, this rigorous selection process might lead to an underrepresentation of clinical diversity in real-world applications. Furthermore, we recognize that the sample size of 99 nodules is relatively modest, potentially limiting the external validity and generalizability of our study results to broader populations. To mitigate these limitations, we recommend that future studies consider including a more diverse patient population, employ multicenter and prospective designs, and apply stratified analysis or other statistical methods to control for confounding variables. These steps would enhance both the sample size and the diversity of participants. In addition, we plan to actively pursue collaborations with other medical institutions in subsequent research efforts, allowing us to validate our models using external datasets and investigate how different types and materials of implants affect ultrasound imaging characteristics.

Our study utilized a retrospective design, which enabled the rapid leveraging of substantial existing clinical and imaging data to provide initial evidence for the application of ultrasound radiomics. However, this approach also introduced certain limitations. Retrospective designs can introduce selection bias because patient inclusion and data collection are based on available records rather than predefined criteria. To mitigate this bias, we implemented several measures, including strictly enforcing inclusion and exclusion criteria, randomly assigning cases to training and validation sets, and anonymizing data to minimize the influence of identifying information. Despite these measures, they cannot entirely eliminate all potential biases. Future research should consider adopting a prospective validation design to address the limitations of retrospective studies. Prospective studies can ensure greater sample representativeness and consistency through clearly defined inclusion and exclusion criteria and standardized data collection protocols, more accurately reflecting real-world clinical scenarios. Moreover, prospective studies are better equipped to capture long-term follow-up data from patients, thus further validating the stability and predictive accuracy of our models.

The nine optimal feature categories we identified included gray-level co-occurrence matrix (GLCM) features, gray-level size-zone matrix (GLSZM) features, shape features, and first-order features. Seven of these feature types were obtained after processing with the wavelet transform filter. Wavelet transforms capture details across different frequencies and directions, a method widely used in image analysis, especially for non-stationary signals, to effectively extract local features and edge information [17]. GLCM features focus on texture patterns, revealing microscopic changes in tissue structure by analyzing the spatial relationships of pixel gray levels in the image. This method has been proven to have good diagnostic performance in various medical image analyses [18]. GLSZM also focuses on texture features, but emphasizes the distribution of region sizes and gray levels, providing quantitative information about tumor heterogeneity. Research indicates that tumor heterogeneity is closely related to its biological behavior and prognosis [19]. Shape features provide information about the geometric characteristics of nodules, parameters often used to assess the invasiveness and growth patterns of tumors. Studies have shown that irregular shapes are associated with an increased risk of malignancy [20]. Additionally, among the first-order features derived after wavelet transform processing, kurtosis, energy, minimum, and maximum are statistical descriptors of the distribution of image gray levels, providing important clues about tissue structure and lesion characteristics for the model [21]. The combination of all these features provides a more nuanced and quantitative analysis than visual inspection alone, aiding in the differentiation between benign and malignant breast nodules.

In our study, the random forest model outperformed other classifiers, such as logistic regression, SVM, and KNN, achieving the highest AUC of 0.79. This superiority can be attributed to several factors. Firstly, random forest’s ensemble learning approach effectively reduces variance and prevents overfitting, which is particularly important in our study’s small-sample setting. Secondly, its ability to model non-linear relationships and handle redundant features allows it to fully exploit the data’s characteristics. In contrast, other models like logistic regression assume linearity, while SVM and MLP require extensive hyperparameter tuning to perform well with limited data. Additionally, random forest leverages bootstrap sampling, maximizing data utilization within the training set and providing robust performance in the validation phase.

Previous studies [22,23] have already confirmed the application value of ultrasound radiomics in diagnosing the benignity and malignancy of breast nodules, but our study is the first to conduct an in-depth exploration specifically for women who had undergone breast augmentation. We found that even in the presence of silicone breast prostheses, ultrasound radiomics can still provide reliable diagnostic information. The AUC value of the constructed random forest model reached 0.787, which is close to the AUC value (0.820) obtained by Romeo V et al. using the random forest algorithm to differentiate between benign and malignant breast lesions [24].

Compared to traditional ultrasound, mammography, and breast MRI, ultrasound radiomics retains key advantages such as being non-radiative, cost-effective, and capable of real-time imaging. Additionally, it enhances diagnostic accuracy and objectivity by integrating advanced image analysis algorithms for the quantitative evaluation of breast nodule features. Traditional ultrasound diagnoses can be subjective and variable, as they heavily rely on the operator’s experience and skills. Mammography excels at detecting microcalcifications, but has lower sensitivity in dense breast tissue and exposes patients to radiation. Breast MRI, while highly detailed, is not ideal for screening due to its high cost and lengthy examination times.

This study reveals the potential application value of ultrasound radiomics in diagnosing breast nodules in women who have undergone breast augmentation. This finding is of great significance for precision medicine because ultrasound radiomics can assist clinicians in more accurately determining the nature of breast nodules, guiding more precise treatment strategies for women who have had breast augmentation. It helps avoid unnecessary invasive examinations or surgeries and promptly identifies malignant tumors to ensure early intervention [25,26].

To further enhance the application of ultrasound radiomics in diagnosing breast nodules after augmentation mammoplasty, we focused on three critical aspects: time efficiency, cost-effectiveness, and training requirements. Ultrasound radiomics can process images within minutes using modern computing resources and automated software, ensuring rapid diagnosis without disrupting healthcare providers’ daily workflows. By leveraging open-source tools, the initial investment costs are minimized, and ongoing research is expected to improve diagnostic accuracy, potentially reducing the need for unnecessary invasive procedures and leading to significant cost savings. Furthermore, a concise and comprehensive training program, complemented by online learning platforms, enables healthcare professionals to quickly acquire the necessary skills. Given that most radiologists are already familiar with ultrasound imaging techniques, this familiarity simplifies the training process and reduces its complexity. By addressing these key areas, we aim to enhance the practicality and accessibility of ultrasound radiomics in clinical settings, thereby improving patient care and diagnostic outcomes.

Regarding future research directions, several avenues can be considered to further advance the field:Conduct multi-center, prospective studies to increase the sample size and diversity, thereby enhancing the generalizability of the research findings;Explore deep learning studies combining multimodal ultrasound images with clinical information and other imaging modalities (such as MRI and CT) to construct more comprehensive diagnostic models;Investigate the temporal trends of ultrasound radiomics features to assess their role in monitoring complications associated with breast prostheses;Gain a deeper understanding of the relationship between ultrasound radiomics features and the type and material of breast prostheses to guide prosthesis selection and long-term management.

## 5. Conclusions

Our study provides preliminary evidence for the application of ultrasound radiomics in diagnosing breast nodules post-augmentation, highlighting the potential of this technology to improve clinical decision-making. However, to achieve widespread clinical application, several challenges need to be overcome, and its efficacy and practicality must be further validated through additional high-quality research.

## Figures and Tables

**Figure 1 curroncol-32-00029-f001:**
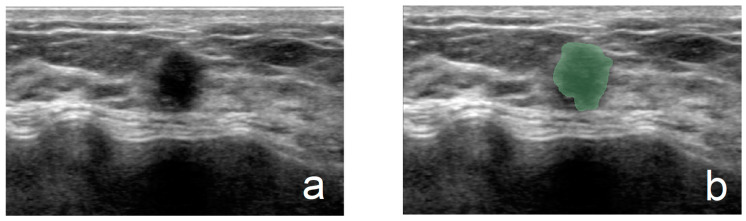
(**a**,**b**) From a 45-year-old patient (fibroadenoma): (**a**) grayscale image and (**b**) ROI of a benign breast nodule. (**c**,**d**) From a 59-year-old patient (invasive ductal carcinoma): (**c**) grayscale image and (**d**) ROI of a malignant breast nodule.

**Figure 2 curroncol-32-00029-f002:**
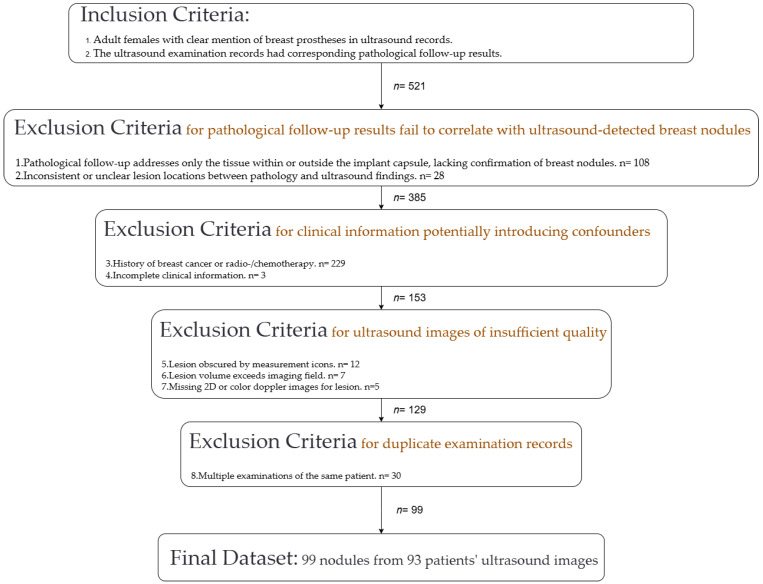
Process of constructing the final dataset, including the inclusion and exclusion criteria applied to select adult women who had undergone breast augmentation for cosmetic purposes.

**Figure 3 curroncol-32-00029-f003:**
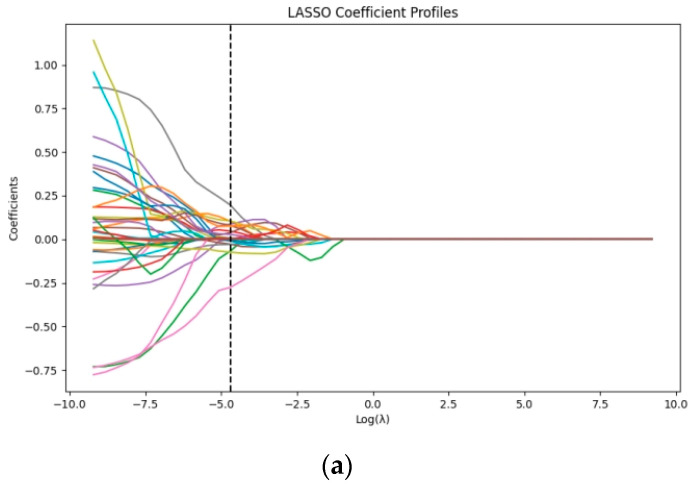
(**a**) LASSO coefficient profiles displaying the variation in coefficients for different variables as a function of the regularization parameter log(λ), showcasing the feature selection process in LASSO regression; (**b**) LASSO parameter-tuning results using 5-fold cross-validation, illustrating the relationship between log(λ) and binomial deviance, with the optimal log(λ) value identified via minimizing the deviance for improved model fitting; (**c**) horizontal bar chart showing the importance of the nine optimal features selected, with longer bars indicating higher feature importance.

**Figure 4 curroncol-32-00029-f004:**
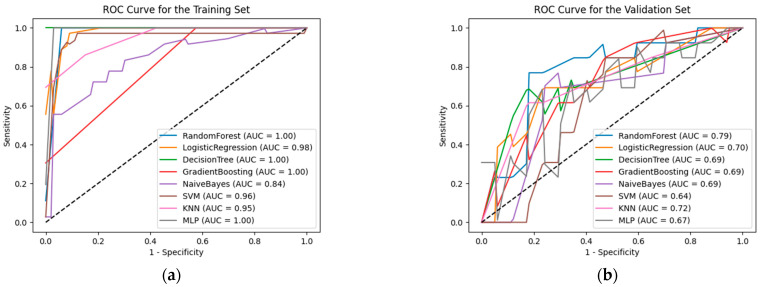
ROC curves of eight models on the training set (**a**) and the validation set (**b**).

**Figure 5 curroncol-32-00029-f005:**
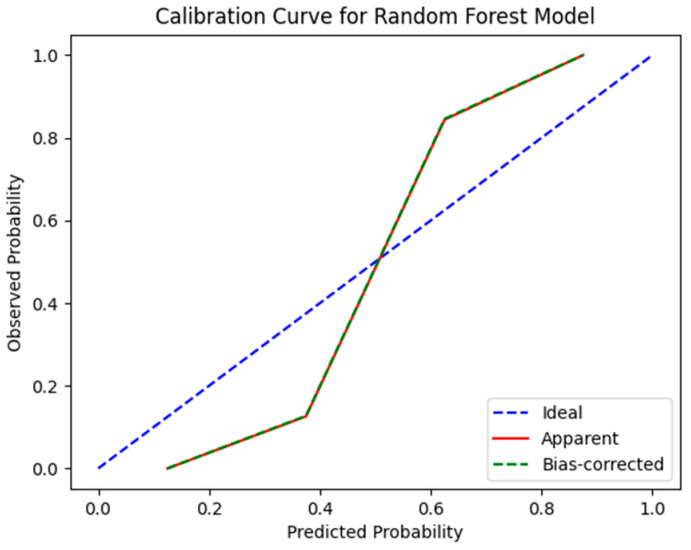
Calibration curve of the random forest model on the validation set.

**Figure 6 curroncol-32-00029-f006:**
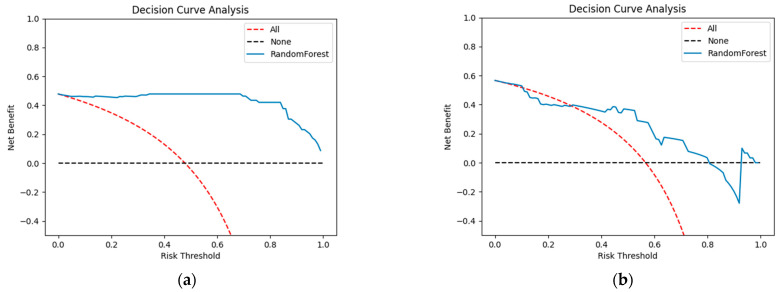
The clinical decision curve (DCA) for the random forest model on the training set (**a**) and the validation set (**b**).

**Table 1 curroncol-32-00029-t001:** Statistics for 99 breast nodules by malignant and benign pathological types.

	Pathological Type	Number of Lesions
Malignant	Invasive ductal carcinoma	44
	Ductal carcinoma in situ	3
	Mucinous carcinoma	2
	Invasive lobular carcinoma	1
Benign	Adenosis	16
	Fibroadenoma	10
	Granuloma	7
	Fibrocystic	6
	Blue gel-like material	5
	Inflammation	4
	Phyllodes tumor, benign	1
Total		99

**Table 2 curroncol-32-00029-t002:** Service years after breast augmentation.

Years Since Surgery	Cases, *n*
≤5	14
6–10	18
11–15	20
16–20	18
>20	10
Information Missing	13
Total	93

**Table 3 curroncol-32-00029-t003:** Distribution comparison of pathology type, lesion location, and maximum diameter between training and validation sets.

	Statistical Test	Training Set (69)						Validation Set (30)						*p* Value
Malignant (M) or Benign (B), *n*	Chi-square test	(M) 32			(B) 37			(M) 18			(B) 12			0.213 (2-sided)
Left (L) or Right (R), *n*	Chi-square test	(L) 36			(R) 33			(L) 17			(R) 13			0.827 (2-sided)
Quadrant, *n*	Fisher’s exact test	c.a. 3	l.i. 9	l.o. 15		u.i. 19	u.o. 23	c.a. 2	l.i. 2	l.o. 5		u.i. 6	u.o. 15	0.554 (2-sided)
Maximum diameter, median ± SD, mm	Mann-Whitney U test	18.77 ± 10.73						16.8 ± 10.08						0.393

**Abbreviations:** c.a., central area; l.i., lower inner; l.o., lower outer; u.i., upper inner; u.o., upper outer.

**Table 4 curroncol-32-00029-t004:** Performance metrics for eight models distinguishing between benign and malignant breast nodules in the training set (TS) and validation set (VS).

Model	Group	Sensitivity	Specificity	AUC (95%CI)	Accuracy	Brier Score	Log Loss Score
Random Forest	TS	1.000	1.000	1.000 (1.000–1.000)	1.000	0.017	0.104
	VS	0.765	0.838	0.787 (0.561–0.960)	0.796	0.197	0.599
Logistic Regression	TS	0.909	0.972	0.977 (0.947–0.998)	0.942	0.058	0.208
	VS	0.529	0.692	0.701 (0.448–0.886)	0.600	0.260	0.830
Decision Tree	TS	1.000	1.000	1.000 (1.000–1.000)	1.000	0.000	0.000
	VS	0.778	0.618	0.698 (0.533–0.871)	0.708	0.292	10.074
Gradient Boosting	TS	1.000	1.000	1.000 (1.000–1.000)	1.000	0.000	0.000
	VS	0.418	0.846	0.692 (0.456–0.900)	0.604	0.355	3.965
Naïve Bayes	TS	0.697	0.806	0.843 (0.754–0.931)	0.754	0.222	0.693
	VS	0.412	0.769	0.692 (0.472–0.919)	0.567	0.248	0.713
Support Vector Machine (SVM)	TS	0.909	0.944	0.959 (0.904–0.998)	0.928	0.080	0.391
	VS	0.529	0.846	0.638 (0.429–0.823)	0.667	0.268	0.948
K-Nearest Neighbor (KNN)	TS	0.849	0.861	0.947 (0.902–0.983)	0.855	0.089	0.247
	VS	0.765	0.615	0.717 (0.493–0.889)	0.700	0.252	5.998
Multilayer Perceptron (MLP)	TS	0.992	0.997	0.998 (0.993–0.999)	0.995	0.004	0.017
	VS	0.581	0.732	0.665 (0.448–0.840)	0.647	0.348	9.833

## Data Availability

The code is available at https://zenodo.org/records/14213723 (accessed on 1 January 2025). Due to patient privacy, the data presented in this study are available on request from the corresponding author.

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
