# Peer review of "Application of Ultrasound Radiomics in Differentiating Benign from Malignant Breast Nodules in Women with Post-Silicone Breast Augmentation"

_curroncol, 2025, doi:10.3390/curroncol32010029_

Round 1

Reviewer 1 Report

Comments and Suggestions for Authors

The paper, titled "Application of Ultrasound Radiomics in Differentiating Benign from Malignant Breast Nodules in Women with Breast Implants," investigates the diagnostic utility of ultrasound radiomics for distinguishing between benign and malignant breast nodules in patients with silicone breast implants. Using a dataset of 99 patients, the authors employed machine learning algorithms to evaluate predictive performance, with Random Forest emerging as the top-performing model. The study emphasizes the potential of radiomics in clinical decision-making, while noting limitations such as sample size and retrospective design. This research contributes to precision medicine by offering a promising diagnostic tool for a unique patient population.

The topic is of interest and addresses a relevant issue in precision medicine and breast cancer diagnostics. However, there are several issues that should be addressed to enhance the scientific rigor, clarity, and clinical applicability of the manuscript.

-         The study uses a relatively small dataset of 99 patients, which limits the generalizability of the findings. The authors should discuss strategies for validation in larger and more diverse cohorts in greater detail.

-         Inclusion and exclusion criteria could be overly restrictive, potentially excluding cases that may be representative of the real-world clinical spectrum.

-       The 7:3 ratio for training and validation could result in underrepresentation of certain subgroups in the validation set. The authors might consider cross-validation techniques for more robust model evaluation.

-       The retrospective design may introduce selection bias. This limitation should be addressed more explicitly, and the authors should outline how prospective validation studies could overcome these biases.

-       While the LASSO method and other feature selection strategies are described, there is limited discussion about why certain machine learning models (e.g., Random Forest) performed better than others.

-       The figures illustrating the Random Forest model’s calibration and ROC curves are helpful but could benefit from clearer labeling and more detailed legends.

-       The flowchart in Figure 1 should be supplemented with more context regarding the decision-making process in inclusion/exclusion.

-       There is limited discussion about how the proposed radiomics tool could be integrated into existing clinical workflows, including time, cost, and training considerations.

-       The authors should compare their radiomics approach to existing diagnostic modalities for this population (e.g., MRI or traditional ultrasound interpretation).

-       Some sentences are overly complex, which can obscure the meaning (e.g., Section 4 Discussion). The authors should simplify and clarify these sections.

-       Occasional grammatical inconsistencies should be addressed during editing.

Author Response

Dear Reviewer,

Thank you very much for taking the time to review our manuscript. We greatly appreciate the constructive comments and suggestions provided, which have significantly enhanced the quality of our work.

Please see the attachment for our detailed point-by-point responses to the reviewers' comments.

In addition to addressing the reviewers' comments, we have also reconsidered and revised certain terminology in the original manuscript to improve clarity and precision. The key changes are as follows:

  1. To avoid ambiguity regarding different methods of breast augmentation, we have revised "breast implant surgery" to "silicon breast augmentation," and "breast implant" to "breast prostheses." The term "implant" is more specific to the surgical placement of a breast prosthesis through an incision. It's important to note that there is another method involving the direct injection of liquid silicone, which has been officially banned due to complications. Therefore, these changes are reflected in the manuscript version marked "TrackedChanges."
  2. In the Results Section(see CleanVersion, line213-215; or see TrackedChangesVersion, line 243-245), we have revised the presentation of statistical outcomes concerning different augmentation methods, clearly distinguishing between implant-based augmentation and injection-based procedures.
  3. In order to distinguish between ultrasound findings and the actual presence of breast nodules, we have refined the use of the terms "lesions" and "nodules." Specifically, "lesions" now refers to the sonographic appearance of detected breast nodules, while "nodules" denotes the objective existence of these structures. These terminological adjustments have been made throughout the manuscript; please see the “TrackedChanges” version for specifics.

We believe that these revisions have improved the precision and clarity of our manuscript. Thank you again for your consideration.

Sincerely,

Buyun Ma

Department of Medical Ultrasound, West China Hospital, Sichuan University, Chengdu 610041, China

maby@scu.edu.cn

Reviewer 2 Report

Comments and Suggestions for Authors

In this paper, the authors tackled a highly interesting research topic – the use of AI-powered algorithms for ultrasound radiomics in breast cancer patients who also have breast augmentation. The field of AI-related research has been expanding exponentially in the last few years.

Although there is a number of studies already done on AI-algorithms used for diagnosing breast cancer, there is none (to my knowledge) done on a such particular population as the one described in the present study (patients with breast cancer AND that are also post-breast augmentation surgery).

The article in well written, with a proper use of academic English language. The study design and methods are well thought up and applied. The results are interesting, well formulated and well interpreted.

Even thou the article is of a high quality there are some minor improvements that should done:

1.      In the phrase starting at line 205: “The statistical results regarding the method of breast augmentation showed that 26 patients underwent injectable augmentation, while 67 patients underwent implant-based augmentation, with silicone being the material used in all cases.” The term “injectable augmentation” should be detailed in the Methods section, and if it means lipofilling than the article title should be changed to “Application of Ultrasound Radiomics in Differentiating Benign from Malignant Breast Nodules in Women post-Breast augmentation procedures” (or something similar)

2.      In paragraph “2.2. Study Subjects” – the terms “exclude cases” is excessively used and should be cut-out / the paragraph should be rephrased.

3.      In line 231, the correct annotation is for Figure nr 3 not nr 2.

Author Response

Dear Reviewer,

We are grateful for your positive feedback and insightful comments on our manuscript. Your recognition of the novelty and importance of our research topic is greatly appreciated.

Please see the attachment for our detailed point-by-point responses to the reviewers' comments.

In addition to addressing your comments, we have made several other important revisions to the manuscript, including enhancing the discussion section, replacing figures, and improving the logical flow of the text:

  1. Added limitations due to the small sample size and retrospective nature of the study, along with strategies for future prospective studies (see TrackedChangesVersion, lines 336-375).
  2. Included a more detailed discussion on why the Random Forest model outperformed other models (see TrackedChangesVersion, lines 397-406).
  3. Addressed comparisons between ultrasound radiomics and other existing diagnostic methods such as traditional ultrasound, mammography, and MRI (see TrackedChangesVersion, lines 415-423).
  4. Expanded the discussion on integrating the proposed ultrasound radiomics tool into clinical workflows, considering aspects like time, cost, and training (see TrackedChangesVersion, lines 431-444).
  5. Replaced Figures 4(a), 4(b), and Figure 5, as the original titles and annotations were unclear. The new figures now clearly illustrate the study results.
  6. Clarified terminology to avoid ambiguity. For example, we specified that "implant" refers to surgically inserted prostheses, so we changed "breast implant surgery" to "silicon breast augmentation" and "breast implant" to "breast prostheses" (see Tracked Changes version).
  7. Simplified complex sentences for better readability. For instance, we modified: 

Original: “Ultrasound radiomics offers promising diagnostic accuracy in the differentiation of benign and malignant breast nodules in the context of silicone breast implants, potentially serving as an additional diagnostic tool for women post-silicone breast implant surgery.” 

Revised: “Ultrasound radiomics demonstrates promising diagnostic accuracy in differentiating benign from malignant breast nodules in women with silicone breast prostheses. This approach has the potential to serve as an additional diagnostic tool for patients following silicone breast augmentation.”

Additionally, we have refined certain phrases to ensure grammatical coherence and logical clarity throughout the manuscript.

Once again, thank you for your valuable feedback. We hope these revisions make the manuscript more scientifically rigorous and clear.

Best regards,

Buyun Ma

Department of Medical Ultrasound, West China Hospital, Sichuan University, Chengdu 610041, China

maby@scu.edu.cn

Round 2

Reviewer 1 Report

Comments and Suggestions for Authors

After addressing all reviewer's comments, the paper has been notably improved and now it is suitable for publication. Thank you. 

Author Response

Dear Reviewer,

We are grateful for your positive feedback and confirmation that the paper has been notably improved and is now suitable for publication. Your thorough review and helpful suggestions have greatly enhanced the quality of our manuscript.

Thank you once again for your valuable contributions.

Best regards,

Buyun Ma

Department of Medical Ultrasound, West China Hospital, Sichuan University, Chengdu 610041, China

maby@scu.edu.cn